# CONDUCTOR: DYNAMICALLY ORCHESTRATING PIPELINE PARALLELISM WITH MULTI-GRANULARITY CONTROL

## ABSTRACT

Pipeline parallelism is a cornerstone of large-scale model training, yet its efficiency is fundamentally limited by straggler-induced pipeline bubbles. This issue is exacerbated by static scheduling approaches, including handcrafted heuristics and Integer Linear Programming (ILP), that are inherently brittle to real-world execution time variance. In this work, we introduce CONDUCTOR, a dynamic, two-tiered scheduling framework that, to our knowledge, is the first to virtually eliminate straggler-induced bubbles under realistic, stochastic conditions. The key insight is to decouple global, long-horizon scheduling from local, instantaneous load balancing. At a **coarse grain**, a reinforcement learning (RL) agent leverages millisecond-scale inference to generate robust global schedules, adapting to runtime dynamics in scenarios where traditional static solvers are intractable. At a **fine grain**, we introduce a dynamic computation migration mechanism that resolves residual micro-bubbles by offloading sub-computations, such as attention heads, from transiently slower to faster devices within a single timestep. Evaluated on large-scale LLM training configurations, our framework outperforms state-of-the-art static scheduling baselines by 5%-14% in throughput and demonstrates superior resilience to injected system noise and execution variance. We believe our results establish a new paradigm for adaptive pipeline scheduling, moving beyond static plans to achieve true zero-straggler performance in practical, large-scale training environments.

## 1 INTRODUCTION

Pipeline parallelism (PP)(Huang et al., 2019; Fan et al., 2021; Narayanan et al., 2019; 2021b) has become a cornerstone for training large-scale deep learning models, enabling computation to scale beyond single-device memory limits by partitioning a model's layers across a fleet of workers . However, its efficiency is frequently undermined by pipeline bubbles, periods of device idleness arising from inter-stage dependencies. A primary cause of these bubbles is the *straggler* problem(Lin et al., 2025), where the entire pipeline stalls, waiting for the slowest stage to complete its computation like Figure3, the difference in computation time across microbatches results in bubbles in pipeline stages . This issue is exacerbated by the strict sequential dependencies inherent in PP, causing any local delay to propagate globally and leading to significant underutilization of computational resources.

Finding an optimal pipeline schedule that minimizes these stragglers is an NP-hard problem (Arató et al., 2005). **Existing static scheduling approaches fall into two main categories, both of which are fundamentally ill-suited for the dynamic nature of real-world execution. The first, handcrafted heuristics**, such as the naive load balancing in GPipe (Huang et al., 2019), are simple to implement but rely heavily on an idealized assumption of uniform computation time. The 1F1B Harlap et al. (2018) scheduling strategy achieves faster memory release by performing the backward propagation at the earliest possible opportunity. This assumption is consistently violated in practice due to a confluence of factors: architectural heterogeneity (e.g., the quadratic complexity of self-attention), the intrinsic asymmetry of Forward/Backward/Weight-Update (F/B/W) passes (Qi et al., 2024; Fan et al., 2021), and system-level stochasticity like network bandwidth fluctuations and kernel performance variance. **The second category, optimization-based methods like Integer Linear Programming (ILP)** (Cai et al., 2020), can theoretically find a static optimum.

However, their practical application is crippled by two fundamental flaws: first, their computational complexity grows exponentially with problem scale, making them intractable for large configurations; second, and more critically, the static "optimal" solution they produce is extremely sensitive to the inevitable, subtle performance perturbations of real-world GPU execution, often rendering it suboptimal in practice. **In essence, all static methods share a common, fatal flaw: they attempt to pre-compute a single, fixed solution for an inherently dynamic and stochastic system, a fundamental mismatch that leads to their brittleness.**

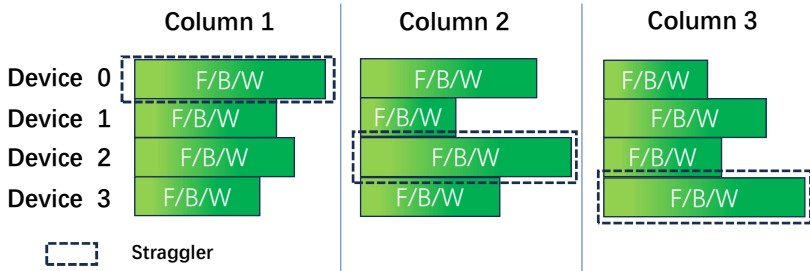

Figure 1: In any given column, the device with the longest execution time becomes the 'straggler' for that step (e.g., Device 2 in Column 1). Because all devices must synchronize at the end of a column before starting the next, this straggler dictates the pace of the entire pipeline, forcing faster devices into idle states and creating performance bottlenecks.

To address these challenges, we argue for a paradigm shift away from static planning towards a strategy of dynamic, real-time control. **Our key insight is that the sources of performance loss occur at different temporal and operational scales, and must therefore be decoupled and addressed with tools of corresponding granularity.** Macro-level pipeline dependencies determine the "base size" of bubbles, while micro-level execution variances exacerbate the problem at every step. Based on this, we introduce CONDUCTOR, a dynamic, multi-granularity control framework.

**At a coarse grain, we employ a reinforcement learning (RL) agent to tackle the macro-level scheduling problem.** The motivation for this choice is that RL learns a *policy*, not a fixed *plan*. Unlike ILP solvers that can take minutes to hours to re-solve, our pre-trained RL policy **leverages its millisecond-scale inference latency to make near-instantaneous scheduling adjustments in response to runtime dynamics**, such as network fluctuations or node preemptions, demonstrating a level of robustness and real-time responsiveness unattainable by static methods.

At a fine grain, we recognize that even an RL policy cannot perfectly predict and negate all minor performance variances caused by hardware or kernel-specific characteristics. To this end, at a fine grain, we recognize that even an RL policy cannot perfectly predict and eliminate all minor performance variances arising from hardware or kernel-specific characteristics. To this end, **we introduce a dynamic computation migration mechanism that, based on profiling results from preceding steps, reactively offloads a small, decomposable portion of computation** (e.g., mlp) from historically slower to historically faster devices. This process is carefully designed such that its communication overhead is effectively overlapped with ongoing local computation, thereby maximally eliminating residual bubbles caused by stragglers without introducing new stalls.

Our primary contributions are:

1. **A Novel Reinforcement Learning Formulation for Pipeline Scheduling Made Tractable via Heuristic-Guided Exploration.** We successfully apply RL to the dynamic pipeline scheduling problem. We overcome the intractably large action space by introducing a two-phase training strategy: (i) Behavioral Cloning on a strong deterministic heuristic (Zero Bubble) to rapidly bootstrap the agent into a high-performance region of the policy space, followed by (ii) online PPO refinement with Domain Randomization. This latter step trains the policy on a distribution of execution timings, learning a robust schedule that is resilient to the system variance that cripples static approaches.

2. **A Fine-Grained, Variance-Reduction Migration Mechanism with a Stall-Free Overlap Guarantee.** We introduce a dynamic computation migration strategy that systemati-

cally eliminates residual stragglers within a single pipeline stage. The mechanism is built on two core technical ideas: (i) a symmetric pairing algorithm that matches the k-th fastest device with the k-th slowest to provably reduce completion time variance, and (ii) a hybrid migration model that intelligently partitions and offloads discrete (Attention Heads) and continuous (MLP blocks) computations. Crucially, we provide a formal analysis in Appendix A that defines the stall-free overlap bound, proving the conditions under which the entire communication overhead of migration is masked by concurrent local computation.

3. **An End-to-End System and Empirical Demonstration.** We built and evaluated the complete CONDUCTOR framework on large-scale GPT-3 model training configurations. Our experiments validate the synergistic relationship between our two tiers: the RL agent creates a "good enough" global schedule that minimizes large-scale bubbles, creating the low-overhead conditions necessary for the fine-grained migration to efficiently polish the schedule to near-zero bubble performance. This combined approach yields throughput improvements of 5-14% over highly optimized static baselines and demonstrates superior robustness in the presence of injected system noise.

## 2 QUANTITATIVE ANALYSIS OF THE PROBLEM

The *straggler* phenomenon(Lin et al., 2025), a primary source of inefficiency in pipeline parallelism, arises from temporal imbalances in stage execution. A quantitative understanding of these imbalances is crucial for developing robust scheduling solutions. This section dissects the key factors contributing to this variance, from architectural determinism to system-level stochasticity.

### 2.1 STAGE-LEVEL COMPUTATIONAL SKEW

In Large Language Models (LLMs), a natural workload imbalance exists between the initial/final stages and the intermediate stages of the pipeline. The first stage (PP Stage 0) performs token embedding, a large table-lookup operation that is often memory-bandwidth bound and is absent from all subsequent stages. Conversely, the final stage (PP Stage N-1) is uniquely tasked with computing the output logits and the cross-entropy loss, a process that includes computationally expensive Softmax operations not performed elsewhere. Although practitioners attempt to balance the pipeline by manually partitioning the Transformer layers, these unique, non-divisible tasks at the pipeline's entry and exit points make achieving perfect temporal equality practically infeasible.

### 2.2 MICROBATCH-LEVEL EXECUTION VARIANCE

Although packing techniques are commonly used in LLM training to ensure each microbatch has a uniform total sequence length, execution time per microbatch can still vary significantly. This variance stems from the architecture of the Transformer block, specifically the self-attention mechanism's computational complexity, which is quadratic with respect to sequence length ($O(T^2d)$)(Vaswani et al., 2017).

Consequently, different combinations of sequences within a packed microbatch will result in different computational loads, even if the total number of tokens is identical. For instance, the computational load for four 4K-length sequences is only 50% of that for two 8K-length sequences. In training scenarios involving long sequences, the portion of FLOPs from the attention mechanism becomes more dominant. When processing a sequence of length $T$ and model dimension $d$, the FLOPs ratio between the attention mechanism and the MLP block can be expressed as:

$$\text{Attention Ratio} = \frac{4Td^2 + 2T^2d}{12Td^2 + 2T^2d}, \quad \text{MLP Ratio} = \frac{8Td^2}{12Td^2 + 2T^2d} \tag{1}$$

This formula illustrates that variations in $T$ directly impact the computational profile, making the execution time of each microbatch difficult to predict accurately. We profiled two combinations with a hidden layer size of 4096 and a sequence length of 8192: one with eight 1K segments and the other with a single 8K segment. Their respective times were 0.569 ms and 3.616 ms, showing a significant practical difference.

## 2.3 INTRINSIC ASYMMETRY OF F/B/W PASSES

Even in an idealized scenario where all other variables are held constant, the core computational passes, Forward (F), Backward-input (B), and Backward-weight (W), have inherent FLOPs discrepancies. As analyzed in prior work(Qi et al., 2024), their execution times typically follow the inequality $T_B > T_F > T_W$. This intrinsic asymmetry is a fundamental source of pipeline bubbles, as any scheduling model that assumes temporal equality among these passes will inherently diverge from the real-world execution profile.

## 2.4 STOCHASTICITY IN COMMUNICATION AND SYSTEM RUNTIME

Beyond deterministic computational factors, non-computational and stochastic elements contribute significantly to pipeline variance. Pipeline parallelism often spans multiple compute nodes, making network latency a critical variable(Wu et al., 2025). Handcrafted schedules often ignore the communication time ($T_{comm}$) required to transfer activations and gradients between stages. In practice, factors such as network bandwidth fluctuations, resource preemption in shared clusters, sudden latency spikes, and varying network topologies introduce unpredictable jitter into $T_{comm}$. This makes the communication time between different device pairs non-uniform and dynamic, posing a substantial challenge for static schedulers.

In summary, these factors do not operate in isolation; their concurrent and compounding effects mean that scheduling models based on idealized assumptions(Narayanan et al., 2021b; Fan et al., 2021) (e.g., assuming F/B/W passes are equal, or even just accounting for their theoretical differences) are insufficient. This discrepancy between the predicted schedule and the actual execution trace leads to an accumulation of pipeline bubbles, exacerbating the straggler phenomenon and significantly hampering the overall efficiency of distributed training.

# 3 DYNAMIC PIPELINE SCHEDULING WITH REINFORCEMENT LEARNING

While handcrafted or ILP-based(Cai et al., 2020) schedules can approach optimality under static assumptions, their prohibitive re-solve latency renders them intractable for dynamic adaptation. Reinforcement Learning (RL)(Murphy, 2025) offers a compelling alternative. The significant computational cost of RL is amortized over a single, offline training phase. Once trained, the policy's millisecond-scale inference latency allows for virtually instantaneous rescheduling with negligible overhead, a capability indispensable for online adaptation. However, learning a policy from scratch in such a large, combinatorial action space remains computationally prohibitive. To address this, we introduce a learning framework that synergizes the strengths of heuristic scheduling with the adaptability of RL.

## 3.1 PROBLEM FORMULATION AS AN MDP

We model the scheduling problem as a Markov Decision Process (MDP), defined by the tuple $(\mathcal{S}, \mathcal{A}, \mathcal{P}, \mathcal{R}, \gamma)$.

**State ($\mathcal{S}$):** A state $s_t \in \mathcal{S}$ at timestep $t$ is a feature vector representing the pipeline's status. It includes: (i) the status of each computational operation (F, B, W for every micro-batch), categorized as pending, ready-to-schedule, executing, or completed; and (ii) the status of each of the $N$ devices, including their current workload and estimated time of availability.

**Action ($\mathcal{A}$):** An action $a_t \in \mathcal{A}$ is the selection of a single operation to schedule from the set of currently available operations, denoted as *ready_operations*. The action space is discrete, and the agent is constrained to select only from this valid set.

**Transition ($\mathcal{P}$):** The transition to the next state $s_{t+1}$ is deterministic, occurring after an operation is scheduled. The state is updated based on the operation's simulated execution time, which determines when the assigned device becomes free and which subsequent operations are added to the *ready_operations* set.

**Reward ($\mathcal{R}$):** We design a multi-objective reward function to guide the agent towards minimizing the total execution time (makespan) while respecting a given memory budget. The reward $R_t$ at each step is a weighted sum of four components:

$$R_t = w_1 R_t^{\text{efficiency}} + w_2 R_t^{\text{balance}} + w_3 R_t^{\text{memory}} + w_4 R_t^{\text{chain}} \qquad (2)$$

where:

- $R^{\text{efficiency}}$ is inversely proportional to the increase in the pipeline's makespan, incentivizing the agent to fill idle gaps.
- $R^{\text{balance}}$ is based on the negative variance of device finish times, directly penalizing schedules that create stragglers.
- $R^{\text{memory}}$ is a penalty term activated when peak memory usage $M_t$ exceeds a soft budget $M_{\text{budget}}$, e.g., $R^{\text{memory}} = -\max(0, M_t - M_{\text{budget}})$.
- $R^{\text{chain}}$ provides a bonus for scheduling operations that unlock a larger number of subsequent dependent operations, prioritizing the critical path.

## 3.2 Policy Learning via Heuristic-Guided Exploration

To make policy learning tractable and effective, we propose a two-phase training strategy that bootstraps the RL agent with expert knowledge.

**Phase 1: Policy Pre-training via Behavioral Cloning.** We first generate a set of high-quality scheduling demonstrations using a strong, deterministic heuristic (e.g., the Zero Bubble schedule). These state-action pairs serve as an "expert policy." The RL agent's network is then pre-trained in a supervised fashion to mimic these expert decisions. This *Behavioral Cloning* (BC) phase rapidly initializes the policy in a promising region of the parameter space, ensuring the agent begins with a competent, low-bubble scheduling strategy.

**Phase 2: Policy Refinement via PPO.** After pre-training, we transition to online reinforcement learning using Proximal Policy Optimization (PPO). The agent now interacts with a simulated environment, using the feedback from the reward function to explore the policy space. Crucially, its exploration is guided from the effective starting point established by BC. This allows the agent to focus its learning on discovering non-obvious scheduling improvements that outperform the original heuristic, particularly in handling the stochasticity that the static heuristic cannot.

## 3.3 Enhancing Policy Robustness via Domain Randomization

A policy trained on deterministic execution times will be brittle. To learn a policy that is robust to real-world variance, we employ *Domain Randomization* during the PPO refinement phase. For each training episode, we parameterize the simulation by sampling the component times $T^P$ and communication times $T^{\text{comm}}$ from distributions derived from the *Live Profiling Phase*, which capture observed mean and variance. Training across this spectrum of simulated environments forces the agent to learn a policy that is not overfitted to one specific timing but is instead resilient to a wide range of potential runtime conditions.

## 3.4 Online Adaptation with Dynamic Triggering

The final, robust policy is deployed for inference during the actual training run. A lightweight monitor compares the expected completion time of each operation against its actual completion time. If the deviation exceeds a predefined threshold, the current real-world state of the pipeline is fed to the trained agent. Leveraging its millisecond-scale inference, the agent instantly generates a new, revised schedule for all remaining operations, a capability made practical and effective by the robustness instilled through our training methodology. Naive scheduling heuristics such as ZB-H1/2 often induce additional pipeline bubbles, as they fail to account for the intrinsic execution time imbalances of the F/B/W passes. Our RL-based scheduler, however, can discover superior scheduling policies, frequently converging to the same optimal schedule as an ILP solver but with significantly lower latency. This ability to rapidly find a high-quality schedule is illustrated in the examples provided in Figures 2, which show a clear reduction in makespan under different situation.

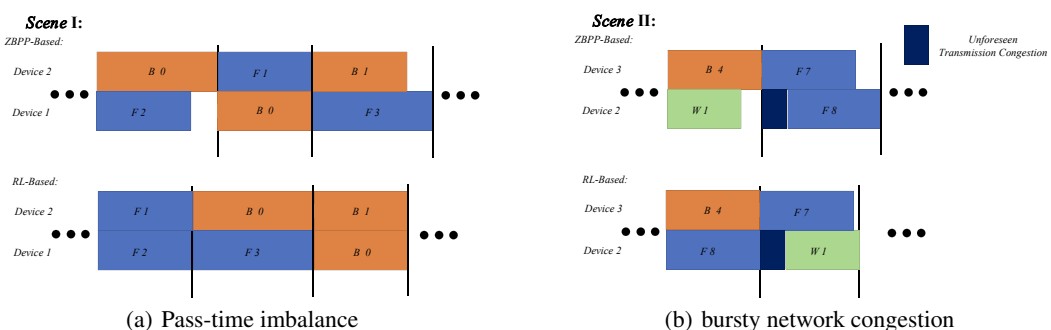

(a) Pass-time imbalance          (b) bursty network congestion

Figure 2: ZBPP vs RL

# 4 DYNAMIC CROSS-DEVICE COMPUTATION BALANCING

While the global policy minimizes makespan, residual inefficiency arises from execution time variance within parallel computation columns. The completion time of any column $t$ for a given pass $P \in \{F, B, W\}$, denoted as $T_{\text{col},t}^P$, is bottlenecked by the straggler device: $T_{\text{col},t}^P = \max_i T_{i,t}^P$. Our objective is to minimize this value by dynamically re-distributing intra-column load.

## 4.1 DECOMPOSABLE COMPUTATION AND PASS-SPECIFIC TIME MODELING

Precise load balancing requires a decomposable model of computation. We partition each primary pass $P$ into its core components: self-attention (Att) and multi-layer perceptron (MLP). The execution time for device $i$ in column $t$ for pass $P$ is modeled as:

$$T_{i,t}^P = (T_{i,t}^P)^{\text{Att}} + (T_{i,t}^P)^{\text{MLP}} + T_{i,t}^{\text{comm}} \tag{3}$$

Component execution times are derived from the *Live Profiling Phase*. A key modeling assumption for the weight gradient pass (W) is that its computation is concentrated in the MLP component, as the attention mechanism has minimal direct parameter gradients. Thus, we assume $(T_{i,t}^W)^{\text{Att}} \approx 0$ (Li et al., 2025).

## 4.2 VARIANCE REDUCTION VIA SYMMETRIC PAIRING

To reduce the maximum execution time, we employ a variance reduction strategy via symmetric pairing. For a given pass $P$, devices are sorted by their execution times, yielding an ordered permutation $\pi$ such that $T_{\pi(1),t}^P \leq \cdots \leq T_{\pi(N),t}^P$.

We form $\lfloor N/2 \rfloor$ pairs by matching the $k$-th fastest device with the $k$-th slowest. For each pair $k = (\pi(k), \pi(N-k+1))$, the target transfer load $T_{\text{transfer}}^{(k),P}$ is the amount of computation required to equalize their execution times:

$$T_{\text{transfer}}^{(k),P} = \frac{T_{\pi(N-k+1),t}^P - T_{\pi(k),t}^P}{2} \tag{4}$$

This target load guides the subsequent migration. As shown in Figure 3, computation load can be migrated between faster and slower devices, and during this process the communication time is completely overlapped (masked) by computation.

## 4.3 HYBRID MIGRATION STRATEGY

The target load $T_{\text{transfer}}^{(k),P}$ is realized by partitioning the underlying computations of the slower device in each pair. The migration strategy differs by pass type.

**For F and B Passes:** We employ a hybrid, prioritized strategy. We first attempt to satisfy the target load by migrating discrete attention heads, which are computationally self-contained. Any remaining load is then satisfied by migrating a continuous fraction of the MLP computation.


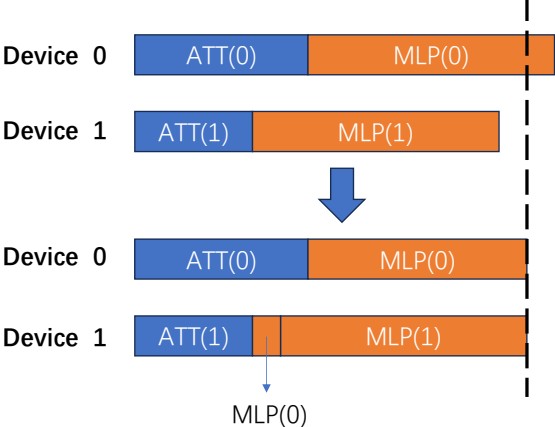

Figure 3: Slower devices offload part of their computation to faster devices

1. *Attention Head Migration:* Let $T_{\text{head}}^{(k),P}$ be the average execution time per attention head on the slower device. We determine the number of integer heads to migrate, $h^{(k),P}$, as:

$$h^{(k),P} = \left\lfloor \frac{T_{\text{transfer}}^{(k),P}}{T_{\text{head}}^{(k),P}} \right\rfloor \tag{5}$$

This is the maximum number of heads that can be migrated without exceeding the target.

2. *Residual MLP Migration:* After migrating heads, a residual transfer load, $T_{\text{rem}}^{(k),P}$, may remain:

$$T_{\text{rem}}^{(k),P} = T_{\text{transfer}}^{(k),P} - h^{(k),P} \cdot T_{\text{head}}^{(k),P} \tag{6}$$

If $T_{\text{rem}}^{(k),P} > 0$, we migrate a fraction of the MLP computation. The MLP splitting ratio, $\alpha^{(k),P}$, is calculated based on this residual load:

$$\alpha^{(k),P} = \min\left(1, \frac{T_{\text{rem}}^{(k),P}}{(T_{\pi(N-k+1),t}^{P})^{\text{MLP}}}\right) \tag{7}$$

**For the W Pass:** Based on the assumption that $(T^W)^{\text{Att}} \approx 0$, the computation is entirely within the MLP component. Therefore, the migration strategy simplifies to only splitting the MLP computation. The splitting ratio $\alpha^{(k),W}$ is calculated directly from the total target load:

$$\alpha^{(k),W} = \min\left(1, \frac{T_{\text{transfer}}^{(k),W}}{(T_{\pi(N-k+1),t}^{W})^{\text{MLP}}}\right) \tag{8}$$

This re-balancing is performed only if the induced communication overhead does not negate the gains. A formal analysis is provided in the Appendix A.

## 5 EXPERIMENTS

### 5.1 SETUP

We demonstrate the superiority of our approach through two methods: first, by simulating whether RL can build an efficient scheduler, and second, by implementing computational balance based on Megatron-LM(Narayanan et al., 2021a), which is tested on GPT-3(Brown et al., 2020). Our experiments utilize up to 16 NVIDIA A100 SXM 80G GPUs distributed across 4 nodes inter connected by a RoCE RDMA network.

Compared methods:

- ZB-1p(Qi et al., 2024): the activation memory limited to pMB(p:stage, MB:the memory of B), which theoretically has the same peak memory as 1F1B.

- ZB-2p: the activation memory limited to 2pMB, which is the least amount of memory to empirically achieve close to zerobubble

- 1F1B and 1F1B-I : 1F1B and interleaved 1F1B methods introduced by(Harlap et al., 2018) and (Narayanan et al., 2021b):

## 5.2 EFFICIENCY OF RL

As illustrated in Figure4, RL-driven dynamic arrangement adds almost no extra idle time, indicating our approach can produce a high-quality schedule in a short period.

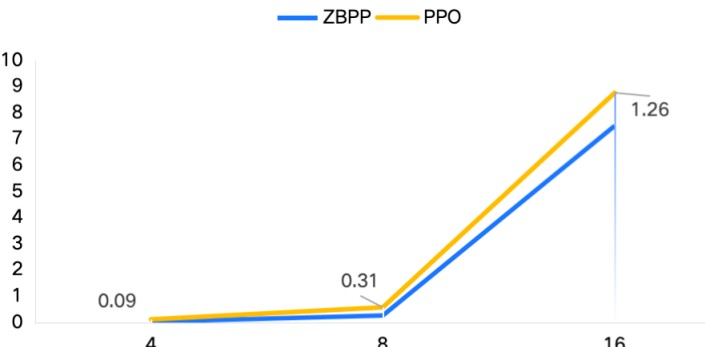

Figure 4: Pure ZBPP ordering (no automation) vs. RL dynamic scheduling.

## 5.3 EFFICIENCY OF COMPUTATION BALANCING

Table 1: Experiment result between previous and ours

| Setup | Model(GPT-3) | 1.5B | 6.2B | 14.6B |
|---|---|---|---|---|
| | #GPU | 8 | 8 | 16 |
| | #Microbatch | 24 | 24 | 48 |
| Samples per GPU per second | ZB-2p/ours | 14.5/15.1 | 4.32/4.51 | 1.81/1.9 |
| | ZB-1p/ours | 12.9/13.4 | 3.88/4.02 | 1.61/1.66 |
| | 1F1B-I/ours | 13.1/13.6 | 4.01/4.16 | 1.54/1.62 |
| | 1F1B/ours | 11.8/13.0 | 3.5/3.78 | 1.4/1.57 |
| Memory (GB) | ZB-2p/ours | 59/59 | 70/70 | 51/51 |
| | ZB-1p/ours | 32/32 | 42/42 | 33/33 |
| | 1F1B-I/ours | 40/40 | 48/48 | 39/39 |
| | 1F1B/ours | 30/30 | 39/39 | 32/32 |

We evaluate the effect of the Chapter 4 computation-balancing scheme by comparing common pipeline-parallel schedules with and without our method, while holding model size, batch size and sequence length fixed. We present the throughput of all methods in Table1. The algorithm targets intra-column runtime heterogeneity: using symmetric pairing and hybrid migration, it redistributes decomposable work across devices in the same column to reduce the column-wise maximum completion time and thereby mitigate tail-induced bubbles.

Let $T_{\max}$ be the column maximum and let $\alpha \in [0, 1]$ denote the fraction of the end-to-end makespan dominated by that column. If balancing reduces $T_{\max}$ by a relative amount $r$, then to first order $\text{makespan}_{\text{new}} \approx \text{makespan}_{\text{old}} (1 - \alpha r)$, so throughput improves roughly by the reciprocal of this factor. Consequently, the attainable speedup depends jointly on the reducible fraction $r$ (set by migration granularity and effectiveness of pairing) and on $\alpha$ (set by the compute/communication split).

Across our benchmark suite and realistic communication costs, the method yields a conservative, reproducible wall-clock improvement of $\approx 5\%$ under typical device variance; when intra-column skew is moderate to high, improvements of $10\% - 20\%$ are common. The net gain is bounded by migration overhead, the minimum decomposable work unit, and memory/ bandwidth constraints; the supplementary material reports break-even points and confidence intervals for these trade-offs.

Next, we conducted a quantitative analysis by selecting different hidden layers and sequence lengths, using the transfer of the MLP component as an example, to examine the benefits brought by our computational balancing. We used a single-layer Transformer as a representative case study to validate the correctness of the communication and overlap mechanisms. The result of linear scaling with the fraction $\alpha = 0.1$ present in Table2.

Table 2: Speedup (%) with hidden dimension ($h$) and sequence length ($L$) on $\alpha = 0.1$.

| Sequence Length ($L$) | $h = 2048$ | $h = 4096$ | $h = 5120$ |
|---|---|---|---|
| 1,024 (1k) | 5.1% | 7.2% | 8.5% |
| 2,048 (2k) | 6.2% | 8.5% | 9.4% |
| 4,096 (4k) | 7.0% | 9.2% | 9.8% |
| 8,192 (8k) | **7.3%** | **9.6%** | **10.2%** |
| 16,384 (16k) | 6.8% | 8.9% | 9.5% |
| 32,768 (32k) | 5.5% | 7.8% | 8.7% |

## 6 CONCLUSION AND DISCUSSION

In this paper, we introduced CONDUCTOR, a dynamic, multi-granularity control framework that addresses the straggler problem in large-scale pipeline-parallel training. Departing from the traditional view of performance variance as purely random jitter, we posit that many latencies arise from periodic, structural bottlenecks. Our approach precisely eliminates these bottlenecks through a **flexible and modular two-tiered strategy**, where each component provides standalone value.

The RL agent replaces the protracted solving process of traditional static methods, finding a high-quality solution to the complex pipeline dependency problem in a fraction of the time. However, we recognize that even such a near-optimal global schedule cannot perfectly nullify the minor performance variances arising from hardware or kernel-specific characteristics. Leveraging millisecond-scale inference, it rapidly generates robust global schedules, effectively handling runtime dynamics that are intractable for traditional solvers like ILP. **Similarly, at a fine grain, our dynamic computation migration mechanism can serve as a general enhancement technique**, applied on top of any high-quality initial schedule to eliminate residual performance imbalances.

However, these two strategies achieve their **maximum efficacy when working in concert**, forming a complete, end-to-end optimization system. They exhibit an ideal complementary relationship: the coarse-grained RL scheduler is responsible for rapidly converging to a "good enough" state at the macro level, ensuring initial pipeline bubbles are sufficiently small to create the ideal, low-overhead conditions for fine-grained migration to operate. In turn, the fine-grained migration handles the low-level system and hardware variances that an RL policy cannot perfectly model, polishing an already excellent schedule towards perfect, "zero-straggler" real-world performance. We believe this modular, yet synergistic, design offers a powerful and effective blueprint for the next generation of adaptive distributed training systems.

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

# A ANALYSIS OF STALL-FREE COMPUTATION MIGRATION (FORWARD PASS)

This section provides a concise formal analysis of the conditions for stall-free migration during the forward pass, a scenario relevant to inference. We model the offloading of a fraction $\alpha$ of a two-layer Transformer MLP from a master rank to a stateless worker, correctly accounting for the full round-trip communication cost.

## A.1 THE STALL-FREE OVERLAP BOUND

For the master rank to avoid stalling, the critical path of the remote task must complete no later than the master's concurrent local computation. This critical path includes sending activations and weights ($C_{\text{send}}$), remote execution ($T_{\text{remote-fwd}}$), and receiving the result ($C_{\text{recv}}$). The stall-free condition is $C_{\text{send}}(\alpha) + T_{\text{remote-fwd}}(\alpha) + C_{\text{recv}} \leq T_{\text{local-fwd}}(\alpha)$.

Assuming linear scaling with the fraction $\alpha$, where $T_{\text{slow-fwd}}$ is the time for the full MLP forward pass on one GPU, this inequality simplifies to the following upper bound for the migratable fraction,

$$\alpha \leq \frac{T_{\text{slow-fwd}} - (C_{\text{send\_act}} + C_{\text{recv}})}{2T_{\text{slow-fwd}} + C_{\text{send\_weights}}} = \frac{\frac{4B_{\text{eff}}}{P} - \frac{2B_{\text{eff}}s}{d_{ffn}B_{\text{net}}}}{\frac{8B_{\text{eff}}}{P} + \frac{2s}{B_{\text{net}}}} \tag{9}$$

This final equation reveals that the maximum stall-free migration fraction is fundamentally determined by the ratio of computational performance ($P$) to network bandwidth ($B_{\text{net}}$), scaled by the geometric properties of the model ($B_{\text{eff}}, d, d_{ffn}$) and the data type size ($s$).

## A.2 CASE STUDY: TWO-LAYER MLP ON AN A100 CLUSTER

We quantify $\alpha_{\max}$ for our target configuration ($d_{\text{model}} = 5120, d_{ffn} = 20480, B_{\text{eff}} = 4096$) on a 100 GB/s network, using BF16 precision and a sustained A100 performance of 312 TFLOP/s.

- **Total Forward Compute Time ($T_{\text{slow-fwd}}$):** The $1.718 \times 10^{15}$ FLOPs of the two-layer MLP forward pass result in a compute time of approximately **5.51** ms.
- **Communication Times:**
  - Activation Send ($C_{\text{send\_act}}$): Transferring the 41.9 MB input activation takes $\approx$ **0.42** ms.
  - Result Receive ($C_{\text{recv}}$): Receiving the 41.9 MB output result also takes $\approx$ **0.42** ms.
  - Full Weights Send ($C_{\text{send\_weights}}$): Transferring the full $2 \times 209.7 = 419.4$ MB remote weight slices (for $\alpha = 1$) would take $\approx$ **4.19** ms.

Substituting these values into Eq. equation 9:

$$\alpha_{\max} \approx \frac{5.51 - (0.42 + 0.42)}{2 \times 5.51 + 4.19} = \frac{4.67}{15.21} \approx \mathbf{0.307}$$

**Conclusion:** Our analysis reveals a significant stall-free migration bound for the forward pass. Up to 30.7% of the MLP's computation can be offloaded to a remote worker with its entire round-trip communication latency being fully masked by the master's concurrent local computation. This rigorously confirms that our fine-grained migration is highly efficient even in compute-lighter scenarios like inference.

