# OpenReview forum: "Conductor: Dynamically Orchestrating Pipeline Parallelism with Multi-Granularity Control"
_ICLR.cc/2026/Conference — ICLR 2026 Conference Withdrawn Submission_

### Official Review · Reviewer_qT58 · 2025-10-23

**Soundness:** 3
**Presentation:** 3
**Contribution:** 3
**Rating:** 6
**Confidence:** 1

**Summary:**

The authors propose CONDUCTOR, a dynamic two-tier framework for orchestrating pipeline parallelism in large-scale model training. CONDUCTOR addresses this by combining a reinforcement learning (RL)–based global scheduler that adapts to runtime dynamics with a fine-grained computation migration mechanism that redistributes workloads across devices to eliminate micro-level stragglers. Experiments on large-scale LLM training configurations show 5–14% throughput gains over baselines and strong resilience to noise.

**Strengths:**

1. This work introduces the first multi-granularity scheduling framework combining RL with fine-grained computation migration.

2. Strong empirical results: consistent 5–14% throughput improvement across multiple LLM scales.

**Weaknesses:**

Evaluation primarily targets GPT-3–scale setups. Expanding the experiments to include a broader range of workloads or additional ablation studies would further strengthen the paper's empirical validity and generalizability.

**Questions:**

I have no questions.

---

### Official Review · Reviewer_ZJRD · 2025-10-27

**Soundness:** 2
**Presentation:** 2
**Contribution:** 1
**Rating:** 2
**Confidence:** 5

**Summary:**

The paper introduces a dynamic scheduling system for pipeline parallelism that aims to reduce straggler-induced pipeline bubbles. The paper does a two level strategy: a) coarse-grained scheduling: a RL scheduler that adapts pipeline execution in real time by selecting which operations to schedule next, and dynamic migration of subcomputations (attention heads and fractional MLP workload) across devices within a single pipeline column to mitigate local execution variance.
Experiments on GPT-3-scale configurations using A100 clusters show 5-14% throughput improvements over static baselines such as ZB-1p, ZB-2p, 1F1B, and interleaved 1F1B, with no increase in activation memory.

**Strengths:**

• Clear articulation of the straggler problem and limitations of static scheduling.
• Communication overlap formally analyzed.

**Weaknesses:**

- The submission claims to be the first to “virtually eliminate straggler-induced bubbles under realistic, stochastic conditions,” yet several dynamic scheduling and adaptive pipeline methods pre-date or coincide with this work (e.g., hybrid-parallel adaptive scheduling, runtime task rebalancing, dynamic intra-pipeline load migration ex "Balanced and Elastic End-to-end Training of Dynamic LLMs, SC'25". These efforts already target the exact same problem scope: online adaptation to runtime variance, dynamic partitioning, and recomputation or shifting of work, reducing bubbles via continuous monitoring and correction
The lack of an explicit comparative baseline weakens the novelty claim and makes it difficult to attribute improvements specifically to the new RL formulation versus established dynamic approaches.
- Table 1 shows only marginal differences versus ZB-2p in several settings. These results suggest that a strong static baseline already stands on its on merit against the proposed approach for for dynamic optimization. The benefit is therefore incremental, not transformative, contradicting strong claims in the abstract. Gains only exceed ~10% over weaker baselines (1F1B), which are not state-of-the-art.
- The RL scheduler is mainly demonstrated in simulation (Fig. 4 lacks quantitative labeling) and not evaluated end-to-end under conditions where RL triggers frequent re-planning. Practical robustness remains uncertain: no results with varying cluster contention, no study of RL inference overhead at scale, and no ablation on reward structure or state encoding
- Communication and Migration Costs Under-Characterized. Despite Appendix analysis, several practical questions are unanswered: how often do migrations occur per iteration?, how do communication paths evolve under congestion?, do repeated migrations saturate inter-node bandwidth? and, impact on overlapping collectives from data parallelism?

**Questions:**

- What are the specific novel contributions beyond prior dynamic pipeline adaptation methods?
- Can you provide an ablation showing the incremental benefit of the RL scheduler vs. using only computation migration?
- Do you have direct bubble-time or idle-time measurements to support the claim of “zero-straggler” performance?
- How often does computation migration occur, and what is the actual communication overhead introduced at runtime?
- How does the approach scale beyond 16 GPUs (when mixed with other parallelism approaches), especially across more complex interconnect topologies? In other words, does the migration interfere with other parallelism modes (tensor/data parallel) in hybrid training setups?
- Can you share key RL training details (policy architecture, reward weighting, convergence behavior)?

---

### Official Review · Reviewer_kyAm · 2025-10-29

**Soundness:** 2
**Presentation:** 2
**Contribution:** 3
**Rating:** 4
**Confidence:** 4

**Summary:**

This paper presents Conductor, a novel pipeline parallelism scheduling mechanism based on a model-driven scheduler. The authors formulate the scheduling problem as a Markov Decision Process (MDP) and employ reinforcement learning (RL) to predict an initial schedule using hybrid rewards. In addition, the paper introduces a dynamic computation migration mechanism, which enables the system to reassign workloads from slower to faster devices. The initial schedule can be further adaptively refined based on real-time profiling when discrepancies arise between the predicted and actual execution timings.

Empirical evaluations demonstrate that Conductor improves end-to-end training time by up to 14%. Furthermore, due to its dynamic scheduling and computation migration capabilities, Conductor effectively reduces residual pipeline bubbles arising from system noise and performance fluctuations.

**Strengths:**

- Addresses an important problem: The paper tackles the challenge of variance in computation times across pipeline parallelism (PP) stages, which can arise from system-level noise or variations in sequence length distributions.
- Novel formulation: The work presents a new problem formulation by modeling the scheduling process as a Markov Decision Process (MDP) and training the scheduler using reinforcement learning (RL). The evaluation results suggest that the proposed model can adapt effectively to unpredictable deviations from the initial schedule.
- Clarity and presentation: The paper is well-written, logically structured, and easy to follow, making the technical ideas accessible and coherent.

**Weaknesses:**

- On design level, A key concern is that the dynamic scheduling mechanism appears to require real-time monitoring of system states during runtime, which may introduce non-trivial overhead. This could be especially problematic in efficient training frameworks, where GPU execution is typically asynchronous.
- Furthermore, since the proposed scheduler seems to operate globally (i.e., controlling all devices), the resulting synchronization cost could be significant.
- Straggler formulation and assumptions: At the methodological level, the formulation of stragglers in pipeline parallelism (PP) is somewhat unclear. The paper assumes a column-based synchronization across stages, which does not align with most existing PP implementations (e.g., 1F1B, 1F1B-I, Zero Bubble). In these schemes, column-level synchronization across all stages does not occur. Instead, each stage typically proceeds independently, constrained only by dependency order (cross devices). The straggler effect described in [1] (the OSDI paper cited) refers to localized dependency delays, not global synchronization stalls. Illustrative examples can be found in implementations such as Zero Bubble Playground (https://sail-zero-bubble-pipeline-parallellism.hf.space/
) and DualPipe (https://github.com/deepseek-ai/DualPipe
), neither of which adopt column-based synchronization.
- Complexity: The migration makes the system design very complicated. Especially considering the concern on column-based synchronization, it might be less important to align all computation in a column to the same run time.
- Evaluation setup: The evaluation setup does not report the scheduler’s overhead, such as scheduling latency, rescheduling frequency, or runtime synchronization cost induced by the global controller. Such data would be valuable for understanding the practical trade-offs of the approach.
- It would be beneficial if the training pipeline or key components of the model-based scheduler could be open-sourced, allowing the community to further validate and extend this line of research.

[1] https://arxiv.org/pdf/2505.05713?

**Questions:**

- Can the authors provide more justification of the column-based synchronization? Considering the concern regarding column-based synchronization, does the proposed method remain effective if this assumption does not hold?
- Could the authors provide additional details about the MDP training process, such as how the policy model is parameterized?
- How frequently is re-scheduling triggered during training, and what factors influence this frequency?
- The paper reports a reduction in residual bubbles in pipeline parallelism. Is there a quantitative analysis of the remaining bubble rate after applying Conductor?

---

### Official Review · Reviewer_n3Fb · 2025-11-01

**Soundness:** 3
**Presentation:** 3
**Contribution:** 3
**Rating:** 4
**Confidence:** 4

**Summary:**

The paper introduces Conductor, an approach to optimize reduce the existance of bubbles in pipelines parallelism during the training phase. Conductor uses RL-based scheduler to choose the policy that reduces the makespan online. The RL is trained offline on two stages; First using  a set of high-quality scheduling examples ( an expert policy), ensuring that the RL-agent starts from a region of the parameter space with low-bubble scheduling strategy; Then, using PPO, they refine the policy with online reinforcement learning on simulated environments. The authors then introduce a new load-balancing method for balancing the load across fast and slow accelerators, by coupling the k-fastest and slowing devices and migrating some of the load while masking computations and communications.

**Strengths:**

The paper introduces a new way to improve pipeline-parallelism, improving the makespan time of training by up to 14%. This translates to huge potential saving for training, increased efficiency, reduced times, and lower energy costs!

The solution looks promising!

**Weaknesses:**

1. The experiments can be better, by training other models besides GPT3. I feel like the whole section is quite thin on both details, and on experiments. Can you also do an ablation study on how each part of your design improves/contributes to the improvement of the makespan?
2. No mention of the cost of training the RL. (time/computations?)
3. Since this is inherently a scheduling paper, one can run simulations for large training with hundreds or even thousands of GPUs. That will also show the scalability of the approach
4. Missing deployment information, e.g., how does this get deployed in a 1000GPU training scenario?

There are a few typos and writing issues:
1. I think you are leaving spaces before the full stops in multiple places.
2. I think movng your Figure 3 a bit up will make readabilitiy better.
3. Some references are wrong, e.g., GPIPE is published in NIPs 2019. You do not say where published. Please go through your references and correct them.
4. Repeated sentence "At a fine grain, we recognize that even an RL policy cannot perfectly predict and negate all minor performance variances caused by hardware or kernel-specific characteristics. To this end, at a fine grain, we recognize that even an RL policy cannot perfectly predict and eliminate all minor performance variances arising from hardware or kernel-specific characteristics."
5. Chapter 4-->Section 4


Suggestions:
1. Some fo the FIgures are occupying a large space and can probably fit next to others
2. Table 1 does not need the memory part since it is practically the same. You can cut that using one sentence in the text/caption and use the space.

**Questions:**

Starting with the Figures:

1. Figure 4 is missing the x/y-axis label. I suppose it is #GPUs/time? Looking at the actual figure, what does 1.26 represent? a 26% increase in time? What would happen is we run the same experiment trying with a larger number of nodes? Is this increase per epoch?
2. Can you please comment on how you see this deployed in a large training with hundreds/thousands of GPUs?
3. Can you comment on how does the different parts of the design affect the performance of your pipeline optimization?

Please also check the weaknesses

---

### Note · Authors · 2025-11-12

I have read and agree with the venue's withdrawal policy on behalf of myself and my co-authors.